

# Interaction estimation of pathogenicity determinant protein βC1 encoded by Cotton leaf curl Multan Betasatellite with *Nicotiana benthamiana* Nuclear Transport Factor 2

Ammara Nasim[1], Muhammad Abdul Rehman Rashid[1], Khadim Hussain[1,2], Ibrahim Mohammed Al-Shahwan[2] and Mohammed Ali Al-Saleh[2]

[1] Bioinformatics and Biotechnology, Government College University Faisalabad, Faisalabad, Punjab, Pakistan
[2] Plant Protection Department, College of Food Sciences and Agriculture Sciences, King Saud University, Riyadh, Saudi Arabia

Corresponding author
Muhammad Abdul Rehman Rashid,
marrashid@gcuf.edu.pk

## ABSTRACT

**Background**. Begomovirus is one of the most devastating pathogens that can cause more than 90% yield loss in various crop plants. The pathogenicity determinant βC1, located on the betasatellite associated with monopartite begomoviruses, alters the host signaling mechanism to enhance the viral disease phenotype by undermining the host immunity. The understanding of its interacting proteins in host plants to develop disease symptoms such as curly leaves, enations, vein swelling, and chlorosis is crucial to enhance the disease resistance in crop plants. The current study was designed to reveal the contribution of βC1 in disease pathogenicity and to unveil potential interacting partners of βC1 protein in the model plant *Nicotiana benthamiana.*

**Methods**. The βC1 gene was cloned in pGKBT7 and used as bait against the cDNA library of *N. benthamiana* and its pathogenesis was tested against the healthy plant and the plants infiltrated with empty vectors. The yeast two-hybrid-based screening was performed to find the interacting factors. Successful interacting proteins were screened and evaluated in various steps and confirmed by sequence analysis. The three-dimensional structure of the Nuclear Transport Factor 2 (NTF2) protein was predicted, and *in-silico* protein-protein interaction was evaluated. Furthermore, protein sequence alignment and molecular phylogenetic analysis were carried out to identify its homologues in other related families. *In-silico* analyses were performed to validate the binding affinity of βC1 protein with NTF2. The 3D model was predicted by using I-TASSER and then analyzed by SWISS MODEL-Workspace, RAMPAGE, and Verify 3D. The interacting amino acid residues of βC1 protein with NTF2 were identified by using PyMOL and Chimera.

**Results**. The agroinfiltrated leaf samples developed severe phenotypic symptoms of virus infection. The yeast-two-hybrid study identified the NTF2 as a strong interacting partner of the βC1. The NTF2 in *Solanaceae* and *Nicotiana* was found to be evolved from the *Brassica* and *Gossypium* species. The *in-silico* interaction studies showed a strong binding affinity with releasing energy value of −730.6 KJ/mol, and the involvement of 10 amino acids from the middle portion towards the C-terminus and five amino acid residues from the middle portion of βC1 to interact with six amino acids of NTF2. The study not only provided an insight into the molecular mechanism of pathogenicity
but also put the foundation stone to develop the resistance genotypes for commercial purposes and food security.

## INTRODUCTION

Plant viruses cause significant damage to the quality and production of crops. They pose a severe risk to food security across the several places worldwide (*Oerke & Dehne, 2004*). Among the plant viruses, *Geminivirus* from the *Geminivirdae* family is one of the largest groups. According to the international committee on taxonomy of viruses (ICTV) classification, the family *Geminivirdae* has evolved to divided into fourteen genera based on genome organization, insect, vectors, and host range (*Fiallo-Olivé et al., 2021*). *Begomovirus* is the largest and economically most important genus of all. Begomoviruses cause more than 90% yield loss in agronomic crops, including cotton and tobacco (*Brown et al., 2015*). Cotton leaf curl Multan virus (CLCuMuV) is a monopartite begomovirus (*Yousaf et al., 2013*). Monopartite begomoviruses are transmitted by whitefly, and their genomes contain one circular, single-stranded DNA (ssDNA) (*Anwar et al., 2020*; *Briddon et al., 2018*; *Mansoor, Amin & RJEov, 2008*). Besides, they contain an additional single-stranded DNA, approximately 1,350 nucleotides in size, known as a betasatellite, involved to promote the disease symptoms. The acquisition of new proteins and the integrating partners are the common causes of viral genome evolution (*Garamszegi, Franzosa & Xia, 2013*). Hence, information about virus-host interacting partners is required to understand the evolution of the viral genome and for disease management.

The betasatellite belongs to the genus *Betasatellite*, consisting of 61 well-recognized species, replicates and spreads through helper begomovirus (*Nawaz-ul Rehman et al., 2021*). All the betasatellites encode a multi-functional pathogenicity determinant, the Beta C1 (βC1) protein, and the βV1 to induce the disease symptoms (*Hu et al., 2020*). The highly conserved βC1 gene, located on the complementary strand of betasatellite, is involved in the reprogramming of host cellular processes to facilitate the begomoviruses-based infections (*Li et al., 2018*). It causes pathogenicity by contributing to disease symptoms such as vein thickening, stunting, leaf deformation, enation, and suppressing gene silencing (*Amin et al., 2011*; *Khan et al., 2017*; *Yang & Zhou, 2017*).

βC1 is reported to interact with many host factors to manipulate the infection and disease symptoms. After infection, βC1 moves to the nucleus of the infected plant cells and interacts with transcription factors to reprogram the gene expression, hence promoting viral propagation and disease symptoms (*Hanley-Bowdoin et al., 2013*). βC1 alters the expression levels of photosystem proteins, resulting a decrease in photosynthesis rate, which is one of the key reasons for developing disease symptoms in plants (*Gnanasekaran et al., 2019*; *Sehrish et al., 2014*). The aforementioned studies have clearly shown that βC1 has been playing a significant role in viral pathogenesis, leading to the initiation of the disease symptoms such as chlorosis, witling, and leaf deformation.

Coordination and regulation of cellular events in a eukaryotic cell depend on the transport of micro- and macro-molecules across the nucleus. Passive transport processes such as diffusion enable micro-molecules (40kDa) to move across the nuclear envelope (NE). The bigger molecules, on the other hand, cannot diffuse through the NE and hence require the assistance of other proteins in their active transport to the nucleus (*Macara, 2001*; *Paschal & Gerace, 1995*). The Nuclear Transport Factor 2 (NTF2) and GTP binding or RAs-related nuclear protein (Ran-proteins) are the two crucial soluble cytosolic factors that assist in active transport through permeabilized cells. To perform this function in nuclear transport, NTF2 interacts with the cytosolic factor Ran. In each transport cycle, Ran delivers one molecule to the cytoplasm which converts to the GTP-bound state and ultimately returns to the nucleus. As Ran is a small protein to diffuse through the nuclear pores, it results in low passive diffusion rate than that required for efficient nucleocytoplasmic traffic. To maintain the asymmetric cellular distribution of regulatory proteins, a small homodimeric protein NTF2 receives the signals and triggers the downstream signaling pathways. It binds specifically the GDP-bound and interacts with nuclear pore complexes (NPC) (*Bayliss et al., 2002*; *Quimby et al., 2000*; *Ribbeck et al., 1998*; *Wong et al., 1997*). Begomoviruses which replicate in the nucleus of the host may exploit this transport system for the movement of the virus genome from cytoplasm to nucleus. Despite the potential biological function suggested by earlier investigations, less was known about NTF2′s contribution to plant defence against pathogen assault. *Nicotiana benthamiana (N. benthamiana)* has attained scientific importance due to its widespread usage as an experimental host plant as its genome data availability and successful inoculation of viral infection are helpful for scientific study (*Goodin et al., 2008*; *Park et al., 2009*). The *N. benthamiana* bears the special ability to be vulnerable to a wide range of plant viruses. As a result, it has emerged as a keystone for virus interaction studies.

The current study was designed to reveal the contribution of βC1 to disease pathogenicity and its interacting partner in the host plant *N. benthamiana*. To identify host factors interacting with CLCuMuB βC1, *N. benthamiana* GAL4-based yeast-two-hybrid (Y2H) approach developed by *Fields & Song (1989)* was used to screen the cDNA library. This protein-protein interaction (PPI) was further evaluated using computational analysis. Identification of β C1 interacting host proteins will broaden the horizon of understanding virus-host interactions required for disease dissemination. Furthermore, the identification of βC1 interacting partners would be helpful in the future to uncover the pathways involved in disease that could be engineered to improve the plants' resilience against ever-evolving viruses.

## MATERIAL AND METHODS

### Sequence retrieval for βC1 and construction of expression cassette

The full-length βC1 gene was amplified from the isolate H181 of CLCuMB using the primers having restriction sites HindIII and BamHI (F: CCAAGCTTATGACAACGAGCG and R: CCGGATCCTTAAACGGTGAACT). The polymerase chain reaction (PCR) profile was adjusted as an initial denaturation at 95 °C for 5 min, denaturation at 95 °C for 30 s,

annealing at 50 °C for 30 s, extension at 72 °C for 30 s and then final extension at 72 °C for 10 min followed by 35 cycles.

The amplified product of βC1 (357 bps) was phenol-chloroform purified and digested to clone under CaMV (Cauliflower Mosaic Virus) 35S promoter of expression vector pJIT-163. Positive clones were confirmed by restriction analysis and further confirmed by standard PCR. Gene expression cassette (1,888 bps) was excised from positive clone samples in pJIT-163 vector using SacI and XhoI. This expression cassette was cloned in binary vector pGreen-0029 at the SacI and XhoI sites.

## Agroinfiltration of expression cassette of βC1 gene

For in-planta interaction evaluation, *N. benthamiana* plants were grown under controlled conditions including 16 h of light, 8 h of dark photoperiod at 28 °C and 25 °C day and night temperature and approximately 65% humidity. Seeds of plants were grown in pots containing peat moss for three weeks.

The expression cassette of βC1 gene and empty vector pGreen-0029 was electroporated into *Agrobacterium tumefaciens* strain GV3101 within electroporation cuvette of (two mm space) according to the previously reported procedure (*Shen & Forde, 1989*). *Agrobacterium* culture nurtured this construct of expression cassette and mock inoculation (empty vector pGreen-0029) was grown within 100 µl of kanamycin and 25 µl of ampicillin per 100 ml media for 48 h (at 28 °C). Bacterial cells were pelleted (5,000×g for 15 min at 4 °C) on an OD at 600 nm of 1 and then resuspended in 10 mM $MgCl_2$ and 100 µM acetosyringone. Cells were left for 2 to 3 h before infiltration at room temperature. The *Agrobacterium* suspension was then infiltrated using 1 ml needle-free disposable syringe on the lower side of three weeks old healthy, young, and fully expanded leaves directly. After infiltration, plants were placed in control conditions of 16 h light photoperiod at 25 °C to 28 °C in an insect-free chamber.

## Generating a bait clone

The bait vector was constructed for yeast-two-hybrid analysis, primers designed having restriction sites NcoI and BamHI (F: 5′AA  CC  ATG  G  AA ATG ACA ACG AGC GGA ACA and R: 5′CC  G  GAT  CC  T TAA ACG GTG AAC TTT TTA). The primers were designed to avoid any codon change and frame-shift mutation. βC1 was amplified using these primers. The conditions for PCR using Phusion High-Fidelity DNA polymerase (New England Biolabs, Ipswich, MA, USA) with the following parameters: 98 °C pre-denaturation temperature for 30 s, denaturation at 98 °C for 10 s: annealing at 55 °C for 30 s: extension at 72 °C for 60 s following 35 cycles. A final extension was 72 °C for 7 min and hold temperature 25 °C. The amplified DNA fragment of βC1 was cloned into the vector pGBKT7 (Clontech Laboratories Inc., Mountain View, CA, USA), It has a GAL4 DNA-binding domain, $Kan^r$ for *Escherichia coli* selection, and *TRP1* for yeast selection. As a result, the gene encoding for βC1 was included in the pGBKT7 vector, which was created as bait for a GAL4-based two-hybrid system and given the name as pGBKT7-βC1.Y2H Gold cells were transformed with 100ng of pGBKT7-βC1 construct along with positive and negative control, whereas, pGBKT7-53 was a positive control and pGBKT7-lam was

used as the negative control. Preparation of master mix by using denatured salmon sperm DNA (ssDNA) at 100 °C for 5 min, 50% PEG and 1M Lithium Acetate (LiAC) was done. The mixture was heated at 30 °C for 30 min, 42 °C for 20 min, and cool the tubes on ice for 10 min. 100 µL of transformants was spread on a single SD/-Trp plate while spreading of positive and negative was done on SD/-Leu/-Trp plates. Bait autoactivation and toxicity test were done by transforming pGBKT7-βC1 construct into Y2H Gold cells along with positive and negative controls.

## Screening of βC1 interacting partner

The *N. benthamiana* cDNA library was generated with an activation domain (AD) having GAL4-AD (Clontech Laboratories Inc., Mountain View, CA, USA). It was generated at Plant Virology Lab at The University of Arizona, USA. Concentrated bait culture of pGBKT7-βC1 working as binding domains (BD) was prepared and combined with library vial AD, after mating the culture was spread on an array of selective agar plates. Initially on 150 mm double dropout SD/–Leu/–Trp (DDO), quarter dropout SD/-Ade/His/Leu/-Trp QDO/X-$\alpha$-Gal agar plates following incubation at 30 °C for 3–5 days.

Mating efficiency was determined by the following formula:

$$\text{Mating efficiency (Diploids \% )} = \frac{\text{Number of cfu/ml of diploids}}{\text{Number of cfu/ml of limiting partner}} \times 100$$

whereas, the number of colony forming units per mL (cfu/mL) of diploid indicated the diploid viability that was equal to number of cfu/mL on SD/–Leu/–Trp, number of cfu/mL of limiting partner was estimated as viability of the prey library that was equal to the number of cfu/mL on SD/–Leu, while the viability of bait was equal to the number of cfu/mL on SD/–Trp.

## Confirmation of positive interaction and rescuing prey plasmid

The true positive prey plasmids were rescued from a single blue colony selected from Y2H experiment and re-streaked on DDO/X-$\alpha$-Gal agar plates (Table 1). The cultures were inoculated overnight in DDO medium at 30 °C. The plasmids were isolated from yeast cells and used for transformation into *E. coli* DH5$\alpha$ cells. Plasmids showed positive interaction was further evaluated and confirmed by transformation and co-transformation.

## Prey plasmid PCR for sequencing

Ten samples showing genuine positive interaction were used for sequence analysis. Plasmids isolated from yeast cells were used for PCR (Table 2). The PCR amplicons were analyzed on 1% Agarose gel. The positive clones were confirmed by amplification with SMART III primer and CDS III primer from both directions (Table 2).

## Sequence homology, alignment, and evolution analyses

NTF2 gene sequence and protein sequences were subjected to the Basic Local alignment search tool for nucleotides (BLASTN) for the identification of their close relatives. Multiple sequence alignments were performed to construct the neighbor-joining phylogenetic tree using MEGA7 (*Kumar, Stecher & Tamura, 2016*) and sequence pairwise identity was analyzed by the Clustal W algorithm in MegAlign application Lasergene package of sequence analysis software (DNA Star Inc., Madison, WI, USA).

**Table 1  Co-transformation scheme of candidate prey with bait and empty bait.**

| Reagents | Selective agar plate | Distinct 2 mm colonies | Color |
|---|---|---|---|
| Candidate Prey + bait | DDO/X-α-Gal | Yes | Blue |
| | QDO/X-α-Gal | Yes | Blue |
| Candidate prey + empty pGBKT7 | DDO/X-α-Gal | Yes | White |
| | QDO/X-α-Gal | No | N/A |

**Table 2  The primers used in the study to amplify the activation domain of protein and the targeted genes for sequencing.**

| Name | Sequence |
|---|---|
| F-AD LD Insert | CTATTCGATGAAGATACCCCACCAAACCC |
| R-AD LD Insert | GTGAACTTGCGGGGGTTTTTCAGTATCTACGATT |
| SMART III primer | AAGCAGTGGTATCAACGCAGAGTGGCCATTATGGCC |
| CDS III primer | TCTAGAGGCCGAGGCGGCCGACATG |

## Protein structure prediction of viral proteins

The protein structure of NTF2 was retrieved from the National Center of Biotechnology Information (NCBI) database (XP_019236968). Evaluation of primary protein structure including the amino acid composition, atomic composition, molecular weight, theoretical isoelectric point (p$I$), estimated half-life, extinction coefficient, instability index, aliphatic index, and grand average of hydropathicity (GRAVY) was determined by using ProtParam (https://web.expasy.org/protparam/) (*Gasteiger et al., 2005*). PSIPRED tools were used for secondary protein structure analysis (http://bioinf.cs.ucl.ac.uk/psipred/). As Protein Data Bank (PDB) did not have a three-dimensional (3D) structure of the NTF2 protein, therefore, I-TASSER online server (https://zhanglab.ccmb.med.umich.edu/I-TASSER/) (*Zhang, 2008*) was used for 3D structure prediction. Then, the model was further refined by using ModRefiner (https://bio.tools/modrefiner). After 3D model prediction and refinement of the NTF2, further structural evaluation, validation, and stereo-chemical analysis were performed. Evaluation of backbone confirmation was done by examining the $\psi/\varphi$ Ramachandran plot acquired from RAMPAGE (https://www.ccp4.ac.uk/html/rampage.html) (*Lovell et al., 2003*). The 3D model was assessed by Verify-3D (https://www.doe-mbi.ucla.edu/verify3d/#:~:text=Determines%20the%20compatibility%20of%20an,the%20results%20to%20good%20structures) to determine three-dimensional protein profiles and compatibility of the atomic model with its own amino acid sequence in comparison to the results of good structure (*Eisenberg, Lüthy & Bowie, 1997*).

## Protein-protein interaction (PPI) evaluation

A web-based server ClusPro 2.0 (https://cluspro.bu.edu), was used to perform Protein-Protein interaction analysis. It performs rigid-body docking of two proteins by giving a basic interface and gives the lowest energy of an interacting complex. ClusPro 2.0 required only two pdb files, receptor, and ligand (*Comeau et al., 2004*). βC1 model was used as a receptor, while predicted NTF2 was used as a ligand. Furthermore, PyMOL v1.7.4 software

and Chimera (*DeLano, 2002*) were used to visualize and analyze the interactions between receptor and ligand complexes, thus obtained.

## RESULTS

### Transient expression of 35S-βC1 from CLCuMB strain in model host plant *Nicotiana benthamiana*

Non-transgenic *N. benthamiana* plants were inoculated with *Agrobacterium* cultures harboring expression cassette of βC1 gene from CLCuMB strain and analyzed the infectivity and symptoms induction ability of 35S-βC1 gene expression cassette. Inoculated leaves sample developed severe phenotypic symptoms of virus infection at 14 days post-inoculation (dpi). The health plants (Fig. 1A) and the plants inoculated with empty vector (Fig. 1B) were compared to the agroinfiltrated plants (Fig. 1C). These symptoms were downward leaf curling and yellowing of inoculated leaves area were observed on the infiltrated leaves (Fig. 1C). With the passage of time, the symptoms became prominent and deeply intense. In all of the four inoculated *N. benthamiana* plants, the leaf showed these typical symptoms at 21–25 dpi when compared to control leaves infiltrated with empty vector pGreen-0029 and healthy plants (Figs. 1A and 1B). The appearance of infectious symptoms in all the nurture leaves confirmed the working efficiency of the 35S-βC1 construct in the model plants and justified the major role of the βC1 gene in diseased symptoms induction in infected plants.

### Yeast two hybrid assay using βC1 as bait and *N. benthamiana* proteins as prey

Based on the available virus genome sequence information (LN886545.1), primers for the βC1 gene were designed. βC1 was cloned into the pGBKT7 vector and confirmed by sequencing showing no point mutation. The results showed 100% identity with the βC1 gene encoded by CLCuMB. Transformation for confirmed clones was done into Y2H Gold competent cells.

A preliminarily study for the autoactivation of βC1 was conducted to investigate the interaction among bait and prey fusion proteins. Yeast cells expressing the reporter genes *HIS3* and *His3* auxotrophic on histidine deficient medium were employed for verification. No growth was observed in yeast cells when pGBKT7-βC1 construct was co-expressed with pGADT7-empty in the histidine-deficient medium. In contrast, positive control showed blue colonies indicating that there was a positive interaction between 53+t, similarly, negative control lam+t showed no interaction at all (Fig. 2).

For mating, the screening of more than $2 \times 10^7$ cells per mL was confirmed by library tittering. Mating efficiency was 5.6% which was sufficient to screen the library successfully (Table 3). Positively interacted proteins showing blue colonies from 150 mm DDO/ X-$\alpha$-Gal plates were further screened and purified by patching on DDO/ X-$\alpha$-Gal plates and QDO/ X-$\alpha$-Gal agar plates (Fig. 3). Positive interacted protein's plasmids were rescued by transformation into *Dh5$\alpha$* of *E. coli*.

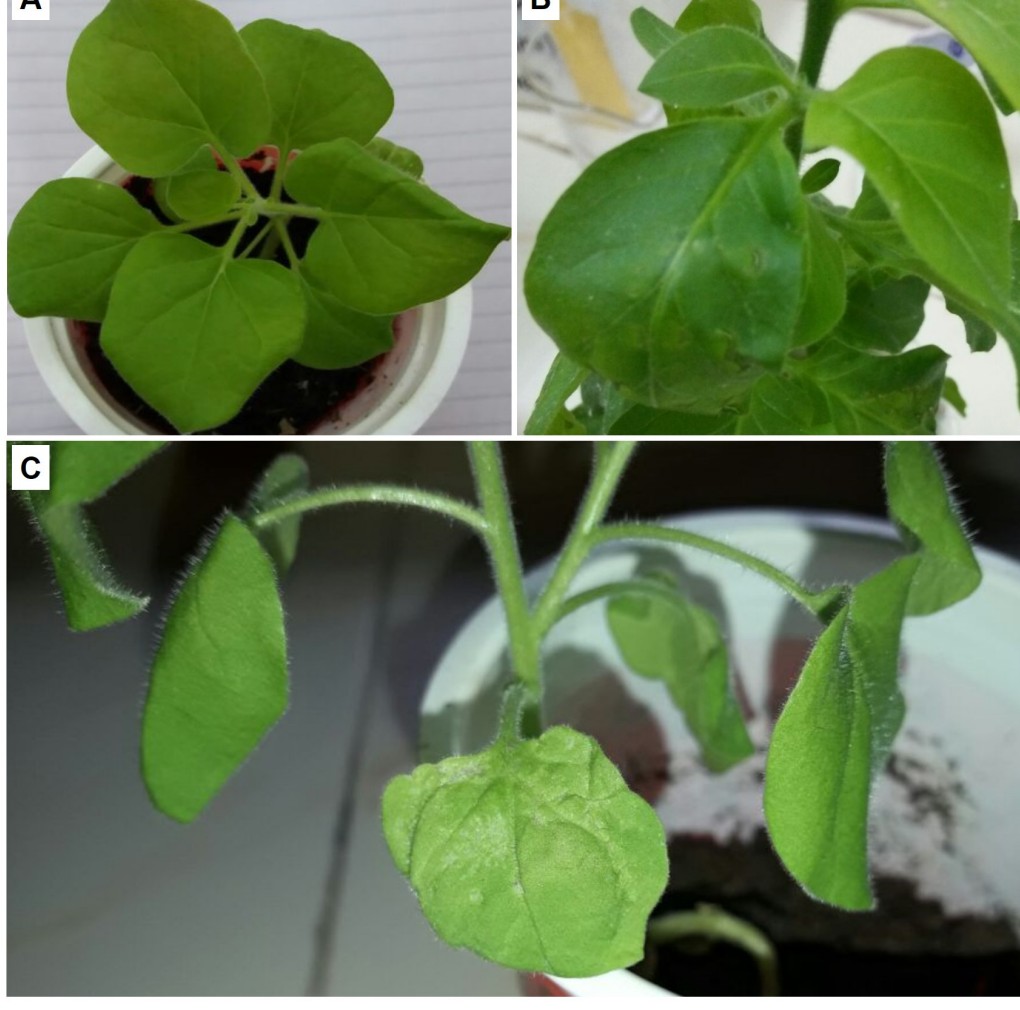

**Figure 1** **The βC1 ORF expression evaluated by agrobacterium-infiltration.** The disease-like symptoms in *Nicotiana benthamiana*. (A) is the healthy plant, (B) leaves infiltrated with empty vector, and (C) the leaves infiltrated leaves with pGreen-βC1.

## Validation of positive interaction through transformation and co-transformation

After being propagated in *E. coli*, the extracted plasmids were again transformed into yeast and co-expressed with the pGBKT7-βC1 construct to examine the reproducibility of the candidate proteins in retransformed yeast cells. On SD/Leu-/Trp-/His-plates, yeast transformants were dispersed, and the survival rate was used to assess the interaction's efficacy. Positive clones were further verified to have strong positive interaction by spreading onto SD/Leu–/Trp–/His–/Ade–plates. As a final point, ten positive clones with strong βC1 interacting proteins were identified.

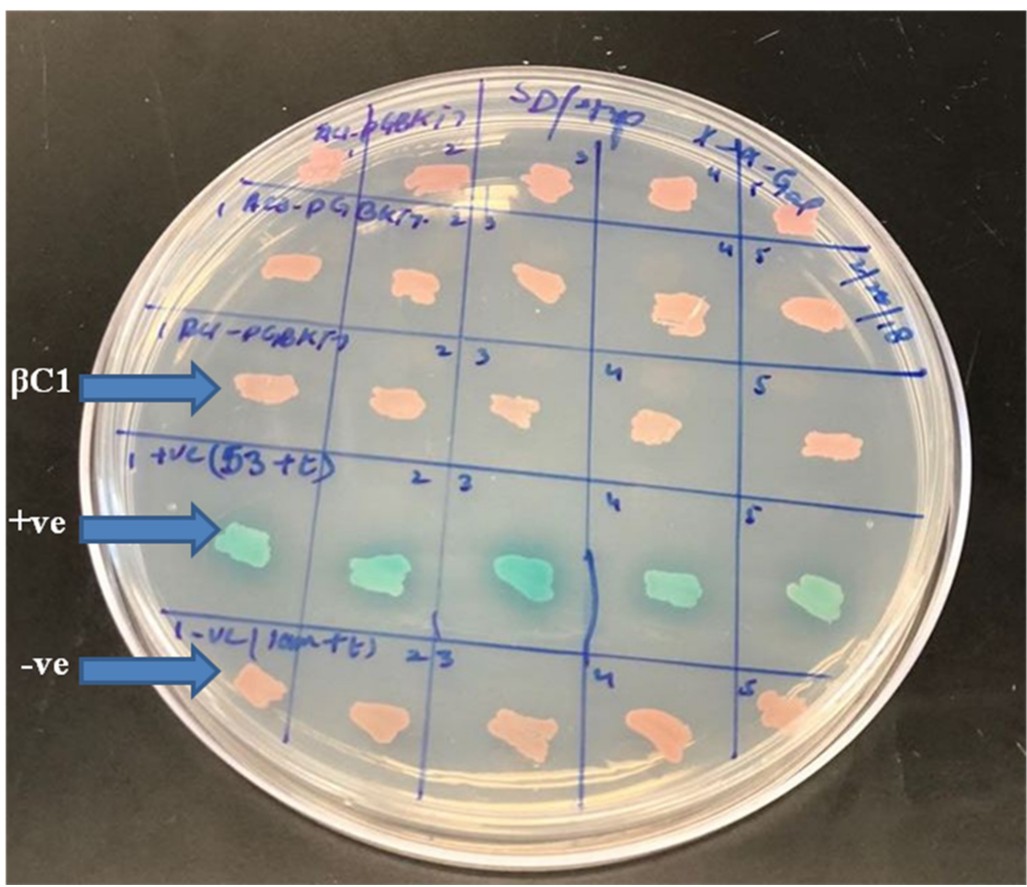

**Figure 2** **For autoactivation test, few random colonies were patched on SD/-Trp X-α-Gal plates.** βC1 and negative control did not show any autoactivation, positive control turns blue indicating the two proteins interacting.

## Identification of *in vivo* βC1 interacting proteins using Y2H

Ten candidate protein plasmids showing genuine positive interaction were sequenced for further analyses (Table S1). The sequencing data was validated for true interaction by homology study, sequence alignment, and open reading frames (ORFs) detection. The sequence homolog was searched in NCBI by the BLASTN tool. All the identified sequences were translated, a significant chunk of sequences was revealed, and ORFs were extracted and aligned. Finally, the similarity index was estimated by the BLAST tool of NCBI. Among the identified sequences, the LOC109217199 mRNA of *Nicotiana tabacum* nuclear transport factor 2-like occurred repeatedly. Therefore, the sequences encoding for Nuclear Transport Factor 2 occurring more than once indicated the best interacting partner of β C1 protein.

## Phylogenetic analysis of NTF2 encoded genes and encoded proteins

Gene sequences of NTF2 from *N. benthamiana* and other plant species were aligned to find its close relatives. Twenty closely related sequences were retrieved from databases with the help of BLASTN. In phylogenetic analysis of *N. benthamiana* NTF2 sequence showed the closest relationship with the members of its own genus *Nicotiana tabacum* and *Nicotiana*

**Table 3** The dilutions spread on agar plates to determine the mating efficiency, and the number of colonies from each plate with dilution factor1:1,000 and 1:10,000.

| Agar plate | Dilution factor | Number of colonies |
| --- | --- | --- |
| SD/-Trp | 1:1,000 | >1,000 |
| | 1:10,000 | 1,000 |
| SD/-Leu | 1:1,000 | 225 |
| | 1:10,000 | 53 |
| SD/–Leu/–Trp (DDO) | 1:1,000 | 18 |
| | 1:10,000 | 3 |

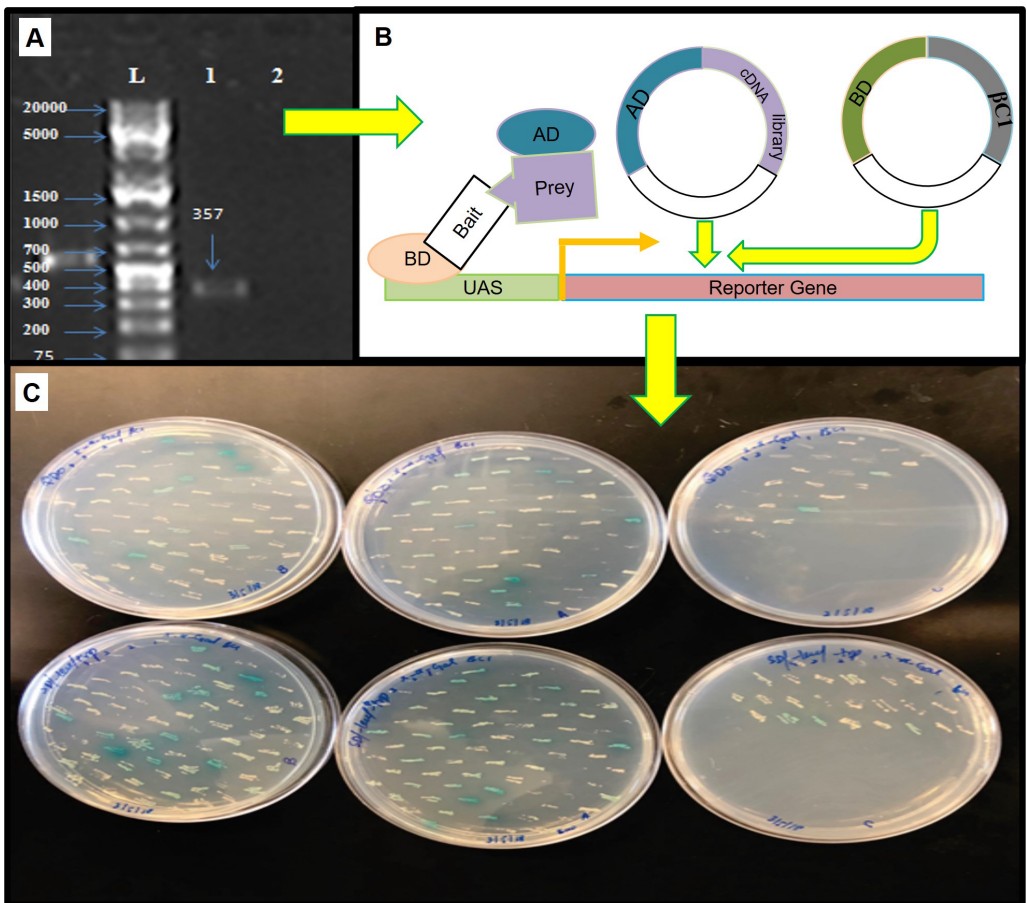

**Figure 3** **The yeast two hybrid (Y2H) screening using βC1 baits with cDNA library of *Nicotiana benthamiana*.** (A) βC1 PCR amplicon on 1% agarose gel, column-1 showing 357 bp band, column-2 showing negative control, column-L has 1kb Plus DNA ladder (Thermo Fisher Scietific, Waltham, MA, USA), (B) Y2H method to identify βC1 interacting proteins, (C) interacting clones (blue colonies) from 150 mm plates patched on DDO/X- $\alpha$-Gal plates and QDO/ X-$\alpha$-Gal plates for the purification.

*attenuata*. Next closest clade of NTF2 gene sequences consisted of *Solanum tuberosum* (potato), *Solanum lycopersicum* (tomato) from the genus *Solanum* of the Solanaceae family (Fig. 4, Table S2).
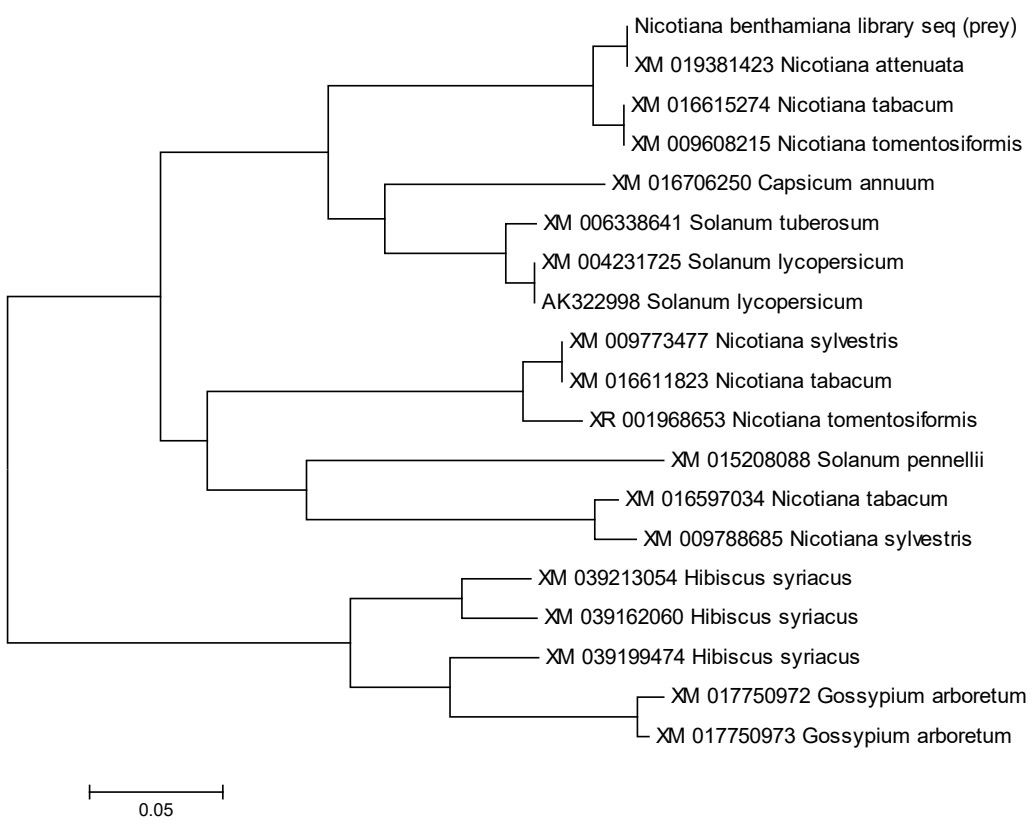

**Figure 4 Phylogenetic analysis of NTF2 gene sequence.** Showing its close relatives.

In phylogenetic analysis, the studied NTF2 protein sequence also showed the closest relationship with same members of its own genus *Nicotiana* as revealed for gene sequence. Further, the NTF2 protein sequence was rooted to the *Brassica rapa* (turnip) and *Arabidopsis* family suggesting the wide pedigree of NTF2 from these species (Fig. 5, Table S3).

### *In silico* prediction of viral protein structure

The NTF2 data retrieved from the online server revealed 123 amino acid residues. Its primary structure showed a molecular weight of 13747.67 Dalton, and a 5.87 p*I*-value (Fig. 6A). The p*I* value was below 7, which indicated the negatively charged protein. Significant stability of protein was revealed by computed instability index of 39.37. The availability of methionine (M) at the N-terminus of the protein, and the negative large average of hydropathicity (−0.089) indicated that the protein was hydrophobic in nature. The glutamine (Q), leucine (L), glycine (G), phenylalanine (F), serine (S), and valine (V) residues were observed in excess (Fig. 6A).

Secondary structure analysis exposed that NTF2 had 20.32% helices and 29.26% extended strands (β-sheets) (Fig. 6B). The predicted 3D model of NTF2 (Fig. 6C) was also evaluated through the Ramachandran plot. The 96.7% of the residues were found in the favored region, 2.5% of the residues were in the allowed region and only 0.8% residues were found in the outlier region (Fig. 6D). I-TASSER was used to predict the 3D structure

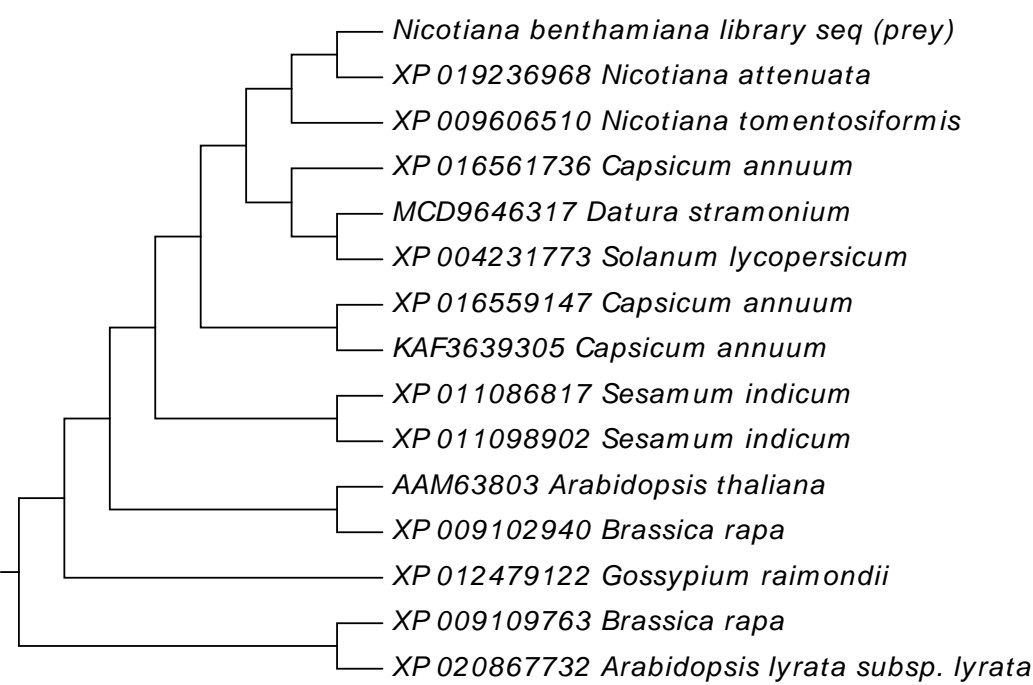

**Figure 5** **Phylogenetic analysis of NTF2 protein.** Showing the close relatives.

of NTF2 and the predicted structure was refined by ModRefiner. Having more than 99% residues of the predicted structure in the favored and allowed regions, the structure was considered to be validated (Fig. 6E).

## Estimation of protein-protein interaction (PPI)

To study the PPI of βC1 and NTF2, βC1 protein was used as a receptor in cluspro while NTF2 was used as the ligand (Fig. 6F). Interaction studies of βC1 with NTF2 showed a strong binding affinity with releasing energy value of −730.6 kJ/mol (Table 4). Analysis of interacting residues showed that 10 amino acids from the middle portion towards the C-terminus and five amino acid residues (TYR48, TYR50, ASP52, ASP58, ASN60) from the middle portion of βC1 interacted with six amino acids of NTF2 (Table 4).

## DISCUSSION

DNA betasatellites associated with helper mono-, and bi-partite begomoviruses depend on the begomoviruses for their replication encapsidation, and spread (*Fiallo-Olivé & Navas-Castillo, 2020*). The satellites contribute to the helper virus's symptomatic range (*Briddon & Stanley, 2006*). Betasatellites consist of a single complementary sense gene (βC1), an adenine-rich region (A-rich region), and a satellite-conserved region (*Briddon et al., 2003*). The βC1, a pathogenicity determinant and suppressor of PTGS, up-regulates viral DNA levels, binds DNA and RNA, and facilitates the virus intracellular movement in plants (*Hu et al., 2020*; *Li et al., 2018*; *Nair et al., 2020*; *Saeed et al., 2007*). β C1 interacts

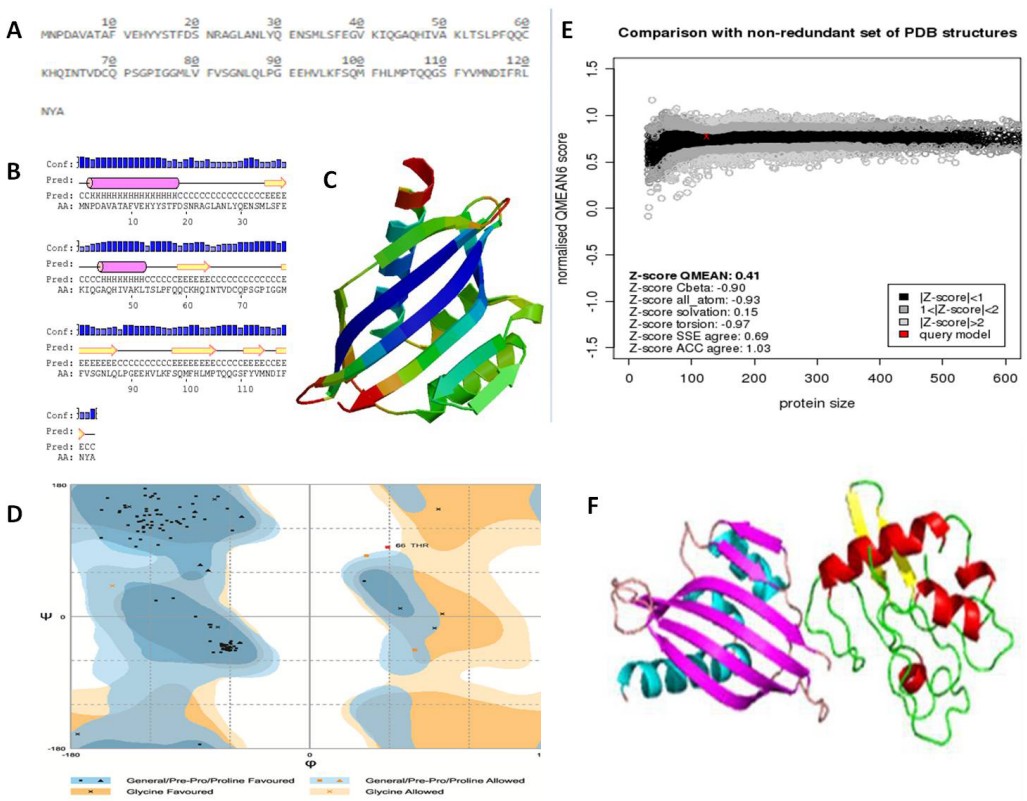

**Figure 6** **Protein primary and secondary structure, and interaction evaluation.** (A) Primary structure analysis of Nuclear Transport Factor 2 protein by PortParam with amino acid composition of protein, (B) secondary structure analysis of Nuclear Transport Factor 2 protein by PRISPRED, (C) tertiary structure of Nuclear Transport Factor 2 protein predicted by I-TASSER, (D) Ramachandran plot, (E) Z-score of NTF2 estimated by Swiss model workspace, (F) molecular docking complex produced in ClusPro online server showing binding affinity of βC1 with Nuclear Transport Factor 2.

**Table 4** **The properties of protein-protein interaction (PPI) including the interacting amino acid residues, and energy released during protein interaction.**

| Property | Description |
|---|---|
| Interacting residues of receptor (βC1 protein) | TYR48, TYR50, ASP52, ASP58, ASN60 |
| Ligand | NTF2 |
| Interacting residues of ligan | GLU38, GLY39, VAL40, PRO56, GLN58, HIS93 |
| Lowest energy | −730.6 kJ/mol |

with the host plant machinery for the successful propagation of disease (*Kamal et al., 2019b*).

During an infection cycle, viruses direct many changes in the intracellular environment of the host cell and provide a safe passage to spread a successful infection by circumventing the antiviral defense mechanism. For the last few decades, virology just focused on the pathogen, but now it is believed that knowing "the host response" is equally or more

important in determining the pathological outcome of infection (*Peng et al., 2009*). Virus-host interaction studies are a helpful tool to understand how viruses manipulate host machinery and cause disease. Biological data retrieval and *in-silico* analyses provide an understanding of new hosts which potentially associated with CLCuD viruses/satellites but lack the information of binding sites (*Kamal et al., 2019a*).

In this study, we identified NTF2 as a strong interacting partner of CLCuMB βC1 by screening a *N. benthamiana* Y2H library with the βC1 as a bait. This interaction was evaluated and validated using the *in-silico* analysis of the protein-protein interaction. Many recent studies used Y2H and transcriptomics approaches to understand the interactional pathways of viruses with host plants (*Kamal et al., 2019a*; *Liu et al., 2014*).

The NTF2 is involved in the nuclear transportation of macromolecules. Mostly the viral proteins or plant viruses interact with the chloroplast proteins to aid viral infection and disease symptoms induction (*DeBlasio et al., 2018*; *Kong et al., 2014*). The NTF2, a protein involved in nuclear trafficking, is conserved in most of the eukaryotes including, yeast, mammals, and *Arabidopsis*. It shows a diverse biological function in different species. Changes in its mRNA expression in various stress conditions depict its involvement not only in normal protein translocation but also in the regulation and mitigation of challenges posed by deleterious conditions (*Quimby et al., 2000*).

This is the first report of the interaction between NTF2 and βC1 protein of the betasatellites. However, other viral proteins have been reported to interact with plant NTF2. For instance, ectopic higher expression of *Arabidopssi NTF2* in *N. benthemiana* results in impaired translocation of a transcription factor and GTPases (*Zhao et al., 2006*). In addition, it expresses at a higher rate in soybean roots challenged with *Meloidogyne javanica* (*Sá et al., 2012*). Furthermore, *ntf2* knockdown wheat lines showed more susceptibility to stripe rust, a disease caused by *Puccinia striiformis,* suggesting a positive role of NTF2 in plant immunity (*Zhang et al., 2018*).

The interaction of NTF2 and βC1 was evidenced using further expression of βC1 in *N. benthamiana*. βC1 was transiently expressed in the *N. benthamiana* leaves. The expression of βC1 induced typical symptoms in the infected agroinfiltrated leaves like chlorosis, leaf witling, and deformation. In host-virus interactions, the offending pathogen frequently develops counter-defense tactics to a host's defensive mechanisms (*Sheikh, 2018*). The presence of upward-curled leaves or even radial leaves in agroinfiltrated plants suggested that this viral pathogenic protein may disrupt the formation of adaxial–abaxial polarity in leaves (*Wang et al., 2015*). The *N. benthamiana* leaves carrying pGreen- βC1, pGreen-empty and healthy-control plants in this study showed different levels of protein expression (Fig. 1). The typical symptoms of leaf deformation were clearly observable in leaves carrying the pGreen-βC1 (Fig. 1C), while there was no expression on leaves with an empty vector and the control-healthy plant (Figs. 1A & 1B). These findings not only validated the integration and proper working of βC1 in *N. benthamiana* but also described the interaction of the viral protein with the host plant machinery to evade host defense mechanism.

Phylogenetic analysis was carried out to evaluate the conservation of NTF2 in different species (Tables S2 and S3). The NTF2 gene of *N. benthaminia* in the current study showed the closest relationship and the highest sequence identity with the NTF2 gene in the *N.*

*tabacum* and *N. attenuata* (Figs. 4 and 5). The previous studies showed that geminiviruses caused different diseases in *Nicotiana tabacum* and *Nicotiana sylvestris* (*Murad et al., 2004*). The background relationship with *Solanum lycopersicum,* and the sequence identity with different members of the genus Gossypium was further elaborated in results. The literature revealed that the members of geminivirus actively target the species of genus Solanum and Gossypium, and induce curly leaf symptoms and leaf deformation (*Naqvi et al., 2017*; *Prajapat, Marwal & Gaur, 2013*; *Varma & Malathi, 2003*). The NTF2 from the *Capsicum annum* showed its conversation with NTF2 of *N. benthaminia*. The yellow leaf curl disease in the genus capsicum has also been reported by begomovirus caused (*Mishra et al., 2020*). High gene and protein sequence identity indicate the NTF2 homologs' presence in a wide range of host plant families. It is involved in disease development by interaction with βC1 protein. The homology and conservation of NTF2 is not only proved in plants but also in yeast and mammals along with elaboration of its function in targeting proteins into the nucleus (*Bhattacharya & Steward, 2002*). Furthermore, it was determined that the sequences of NTF2 in yeast, humans, and Arabidopsis were noticeably similar (*Zhao et al., 2006*).

*In silico* protein-protein docking results of this study depicted the strong interaction between NTF2 and viral protein CLCuMB βC1, which was further substantiated by Y2H assays. Overall, the current study found that NTF2 had a strong binding affinity with βC1 as indicated by the involvement of six residues with the lowest energy value of −730.6 kJ/mol. So, suggesting its potential role in nuclear trafficking of begomovirus genome, its replication, and dissemination in neighboring cells can cause chlorosis, leaf curling, and/or witling symptoms. Various multi-pronged *in-silico* techniques on the basis of energetic, sequence conservation analysis, and the binding site might be useful to envisage the host-pathogen interaction such as geminivirus proteins and their interacting partners in cotton (*Kamal et al., 2019a*).

## CONCLUSION

This study aimed to evaluate the pathogenicity of βC1 protein and its interacting partners. The agroinfiltrated leaves of *N. benthamiana* showed the disease symptoms as chlorosis and leaf deformation. We proposed a new role for β C1 in plant immunity responses that has been in veil until now. The Y2H-based experiment and *in-silico* protein analyses presented in the current study suggested that the βC1 strongly interacts with NTF2, for nuclear transportation. Further studies on βC1 tagging with the green fluorescent protein (GFP) in wild-type and *ntf2* mutant background will shed light on the NTF2 requirement for its nuclear translocation and disease propagation. In the future, the interaction site mapping and mutant studies will aid in finding the NTF2 domain involved in interaction with βC1. Finding interaction sites will pave the way to find the common mechanism involved in βC1 nuclear translocation in many host plants. The results in this study may not only provide an insight into resistance mechanism but also set a base for future research on the involvement of cytoplasmic nuclear transportation in protein interaction and disease defense mechanisms.

## ACKNOWLEDGEMENTS

We would like to acknowledge the School of Plant Sciences, University of Arizona, USA for their assistance in providing the *N. benthamiana* "Mate and Plate" library.

### Funding

This work was funded by the National Plan for Science, Technology, and Innovation (MAARIFAH), King Abdul-Aziz City for Science and Technology, Kingdom of Saudi Arabia, grant number (2-17-04-001-0008). The funders had no role in study design, data collection and analysis, decision to publish, or preparation of the manuscript.

### Grant Disclosures

The following grant information was disclosed by the authors:
The National Plan for Science, Technology, and Innovation (MAARIFAH).
King Abdul-Aziz City for Science and Technology, Kingdom of Saudi Arabia: 2-17-04-001-0008.

### Competing Interests

The authors declare there are no competing interests.

### Author Contributions

- Ammara Nasim conceived and designed the experiments, performed the experiments, analyzed the data, prepared figures and/or tables, authored or reviewed drafts of the article, and approved the final draft.
- Muhammad Abdul Rehman Rashid conceived and designed the experiments, performed the experiments, analyzed the data, prepared figures and/or tables, authored or reviewed drafts of the article, and approved the final draft.
- Khadim Hussain conceived and designed the experiments, performed the experiments, analyzed the data, prepared figures and/or tables, authored or reviewed drafts of the article, and approved the final draft.
- Ibrahim Mohammed Al-Shahwan conceived and designed the experiments, authored or reviewed drafts of the article, and approved the final draft.
- Mohammed Ali Al-Saleh conceived and designed the experiments, authored or reviewed drafts of the article, and approved the final draft.

### Data Availability

Sequence data is available in the Supplemental Files and at GenBank: XM_019381423, XM_016615274, XM_009608215, XM_006338641, XM_016706250, XM_004231725, AK322998, XM_009773477, XM_016611823, XR_001968653, XM_016597034, XM_017750972, XM_017750973, XM_009788685, XM_015208088, XM_039213054, XM_039199474, XM_039162060.

## Supplemental Information

Supplemental information for this article can be found online at http://dx.doi.org/10.7717/peerj.14281#supplemental-information.

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
