# Peer review of "Interaction estimation of pathogenicity determinant protein βC1 encoded by Cotton leaf curl Multan Betasatellite with Nicotiana benthamiana Nuclear Transport Factor 2"

_PeerJ, doi:10.7717/peerj.14281_

## Round 0.1 · original submission · Major Revisions

Kindly revise manuscript as per referee comments.

Regards Dr Zia

Reviewer 1 ·

Basic reporting

The quality of used English language is of insufficient quality.

Several irrelevant references are used and several important and relevant references are not cited.

Figures are of poor quality. Figure legends are not properly described.

The authors reached a conclusion that goes far beyond the scope of the study. Furthermore, the hypothesis is missing, and it should be stated clearly in the introduction.

Experimental design

Yes.

Research question can be improved.

Method sectioncan be improved and need to demonstrate all the things (see my comments)

Validity of the findings

See my comments

Additional comments

The manuscript is aimed to investigate to unveil the potential interacting partners of βC1 protein in the model plant Nicotiana benthamiana. The investigation is interesting, giving the new insight into host-virus interaction. The script was planned to provide valuable information to develop resistance mechanism against betasatellite requiring geminiviruses. Nonetheless, the script has some flaws that necessitate a thorough review and revision of the text. The authors reached a conclusion that goes far beyond the scope of the study. Furthermore, the hypothesis is missing, and it should be stated clearly in the introduction.
 Main concerns refer to the presentation of results, the figures and the language.
 Authors are non-consistent while using the microbe names, NTF2 gene name and abbreviations.

For detailed comments - see the appended file.

Annotated reviews are not available for download in order to protect the identity of reviewers who chose to remain anonymous.

Reviewer 2 ·

Basic reporting

This manuscript is well written. English is relatively good and raw data is provided.

Experimental design

Research question is well defined and proven through experiments.

Validity of the findings

I have one reservation though. There is no interaction studies in-vivo. I would recommend authors should perform an experiment where they can show in vivo interaction of betaC1 and nuclear transport factor 2.

---

## Round 0.2 · Minor Revisions

Dear Sir

I suggest to accept this paper if authors do the minor revisions suggested by referee

Reviewer 1 ·

Basic reporting

The script requires a thorough revision of the English language. Several references are irrelevant.

Experimental design

no comment

Validity of the findings

no comment

Additional comments

General Comment:

The script has been substantially revised by the authors, but it still requires a lot more improvements, as follows;

• It requires a thorough revision of the English language.
• Several references are irrelevant.
• The modified title of the script is not very convincing. Authors may consider about changing it.
• Fig. 1; the inoculated plants resemble the mock- and non-inoculated plants. I am not convinced with the authors’ assertion. This experiment should be revised.
• With respect to Fig 4, authors wrote “the studied NTF2 protein sequence also showed closest relationship with same members of its own genus Nicotiana tabacum and Nicotiana sylvestris as revealed for gene sequence.” However, Fig. 4 depicts that prey sequence is constituting a close group with “Nicotiana attenuate”. BTW, it should be N. attenuata.
• Additionally, Fig. 4; authors claimed that “They looked to be evolved from the Gossypium raimondii.” Again this isn’t true.
• The used sentence in discussion section “DNA betasatellites associated with helper mono-, bi-partite begomoviruses replicate, encapsidate, and spread via begomoviruses” is highly vague and need to be corrected.
• Supplementary data is not referred anywhere in the script.
• The study demonstrated by Kamal et al., 2019 (https://www.frontiersin.org/articles/10.3389/fpls.2019.00656/full) should be discussed with the results of the study.

---

## Round 0.3 · accepted · Accept

The authors have addressed all concerns and now the paper may be accepted.